# BIG.LITTLE VISION TRANSFORMER FOR EFFICIENT VISUAL RECOGNITION

## ABSTRACT

In this paper, we introduce the big.LITTLE Vision Transformer, an innovative architecture aimed at achieving efficient visual recognition. This dual-transformer system is composed of two distinct blocks: the *big* performance block, characterized by its high capacity and substantial computational demands, and the *LITTLE* efficiency block, designed for speed with lower capacity. The key innovation of our approach lies in its dynamic inference mechanism. When processing an image, our system determines the importance of each token and allocates them accordingly: essential tokens are processed by the high-performance big model, while less critical tokens are handled by the more efficient little model. This selective processing significantly reduces computational load without sacrificing the overall performance of the model, as it ensures that detailed analysis is reserved for the most important information. To validate the effectiveness of our big.LITTLE Vision Transformer, we conducted comprehensive experiments on image classification and segment anything task. Our results demonstrate that the big.LITTLE architecture not only maintains high accuracy but also achieves substantial computational savings. Specifically, our approach enables the efficient handling of large-scale visual recognition tasks by dynamically balancing the trade-offs between performance and efficiency. The success of our method underscores the potential of hybrid models in optimizing both computation and performance in visual recognition tasks, paving the way for more practical and scalable deployment of advanced neural networks in real-world applications.

## 1 INTRODUCTION

Vision Transformer (ViT) (Dosovitskiy et al., 2020) has increasingly influenced the field of computer vision since its introduction. It demonstrates exceptional performance in fundamental tasks such as image classification (Deng et al., 2009), image segmentation (Kirillov et al., 2023), and object detection (Li et al., 2022). Furthermore, the flexibility of the transformer architecture enables ViT to act as a crucial conduit between visual and linguistic information in multimodal models (Liu et al., 2023a; Chen et al., 2023), significantly contributing to their rapid development. Additionally, due to the scalability of ViT, as the model sizes increase, ViT is able to effectively learn richer representations of images. Therefore, making large ViT is highly desirable for downstream tasks and applications.

Despite the impressive performance of ViT, its slow inference speed remains a notable drawback. For instance, models utilizing ViT-Huge with more than 600M parameters as a core component, such as the Segment Anything Model (SAM) (Kirillov et al., 2023), may operate at less than 2 FPS on a high-end NVIDIA A100 GPU (Xiong et al., 2023), not to mention ViTs with billion-level parameters (Zhai et al., 2022; Sun et al., 2023; Dehghani et al., 2023; Chen et al., 2023). This limitation significantly hinders the practical deployment of ViT-based models in real-world applications and there is an urgent need for improving the inference speed of ViT models.

To tackle this issue, a variety of strategies have been developed to enhance the inference speed of ViT in recent years. Some works address the problem from the model perspective, either by distilling the knowledge into a lighter-weight model (Xiong et al., 2023), or lowering the precision of model parameters (Dettmers et al., 2022). Instead, inspired by the discovery that only representative tokens are crucial for the final prediction, token pruning methods emerge and speed up the inference by

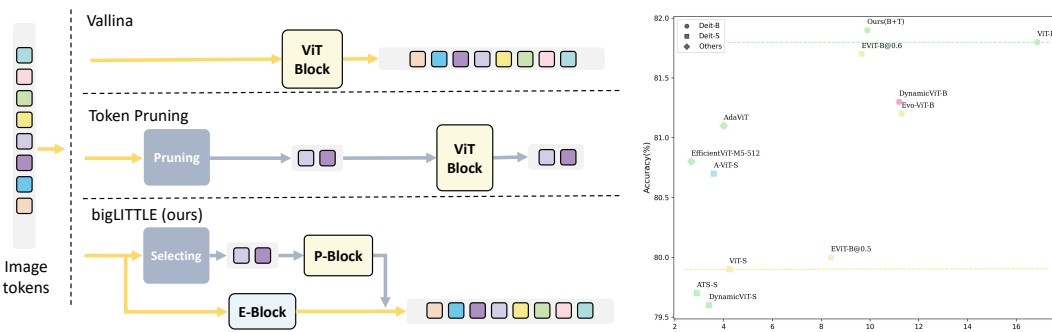

Figure 1: **Comparison between big.LITTLE and conventional token pruning and Performance of various token pruning strategies.** The left diagram compares the standard ViT, token pruning which selectively removes less important tokens, and big.LITTLE ViT that integrates both high-capacity performance blocks (P-Block) and high-efficiency blocks (E-Block) for dynamic token processing. The right demonstrates the performance and efficiency of different models and our big.LITTLE ViT on the ImageNet classification task. Here, shape represents the baseline corresponding to the model. This visual comparison underscores the ability of big.LITTLE ViT to maintain high accuracy while significantly enhancing processing speed.

reducing the number of tokens layer by layer (Xu et al., 2022; Liang et al., 2022). Although they have shown promising results with the enhanced model speed on the image classification task, which only requires predicting one class label for each image, directly dropping the unrepresentative tokens can disrupt the spatial structure of image tokens and lose the context information. Such incomplete information flow may lead to sub-optimal model performance when performing downstream perception tasks, such as image segmentation.

Therefore, to achieve higher inference speed while preserving the context information images, we recognize that all tokens are needed, but not all tokens are equally important. Intuitively, we humans have a large field of view, but will only focus on a small area each time when we see the world. For the focused area, we pay more attention to detailed processing while keeping an eye on the surroundings.

Motivated by this observation, we introduce a novel system called big.LITTLE Vision Transformer (bLViT), which comprises *big* performance blocks and *LITTLE* efficiency blocks within the ViT architecture. In our design, only a few important tokens are updated with the performance blocks each time, which ensures the **performance** of the model during the inference with a reduced computation. For the less important areas, we keep the context information but pay less computation cost to enable high inference **efficiency** with the efficiency blocks. Although most image tokens are pruned from the performance blocks based on their importance, the efficiency blocks ensure that all tokens continue to update layer by layer, preserving the structured attributes of image tokens. Whether a token is processed by the big model is determined by its importance score from prediction layers. Throughout training, our differential design on token selection enables the prediction layers to appropriately route critical tokens to the performance blocks, ensuring intensive computation for those deemed most significant.

We demonstrate the efficacy of our bLViT through applications in image classification and image segmentation tasks, employing DeiT (Touvron et al., 2021) and SAM (Kirillov et al., 2023) as the base models within our big.LITTLE system. The experimental results exhibit a competitive trade-off between computational speed and accuracy, highlighting our model's capability to effectively balance performance and efficiency.

To summarize, our contributions are as follows:

1. We propose a big.LITTLE Vision Transformer (bLViT) model which effectively prunes tokens to reduce computational overhead while preserving the context information and achieve a better speed-accuracy tradeoff.

2. We conduct experiments on image classification and image segmentation tasks and demonstrate the efficacy and efficiency of our bLViT.

3. We perform extensive ablation studies to verify the design choice of our models and improve its performance. We hope these designs could benefit the future development of such heterogeneous model architecture.

## 2 RELATED WORK

**Vision Transformer.** Vision Transformer (Dosovitskiy et al., 2020) has achieved a great success and shows state-of-the-art performance on many tasks including image classification (Touvron et al., 2021), object detection (Li et al., 2022), semantic segmentation (Strudel et al., 2021; Cheng et al., 2022; 2021), etc. The long-range dependency modulation enables its capability to encode rich contextual information, which can benefit downstream tasks by providing better image representations. Therefore, a stream of work studies how to adapt plain ViT to different tasks to optimal the network architecture and boost the performance (Wang et al., 2022; Zhang et al., 2022a; Yao et al., 2024), using the pretrained model on large-scale datasets with different pretraining strategies (Zhang et al., 2022b; Oquab et al., 2023). Despite its wide application and high performance, the computational burden poses challenges to the inference speed and practical deployment in resource-constrained environments. A better speed-accuracy tradeoff for the model is desirable.

**Computation Reduction.** To reduce the computation of existing models, several works have attempted to prune the input tokens (Liang et al., 2022; Xu et al., 2022; Rao et al., 2021) or merge the input tokens (Marin et al., 2021; Bolya et al., 2022). This is achieved by identifying and retaining only the most informative tokens, effectively reducing the number of tokens to process. AdaViT (Meng et al., 2022) further tries to partially or entirely drop the layers for all tokens. This type of method can achieve good speedup with only marginal performance decreases on ImageNet classification. However, few of them have proven the model can work with downstream tasks besides image classification as many tokens are dropped in a very early stage.

**Speedup with Small Model.** Leveraging a smaller model is another way to speed up model inference. The speculative decoding framework (Kim et al., 2023) introduces a mechanism using a separate large language model along with a smaller one to improve inference speed in natural language processing. Big-little Net (Chen et al., 2018) proposes to learn multi-scale feature representations with Big-Branches process the low-resolution input and Little-Branches process the high-resolution input to balance the computation on image and speech recognition. Mixture-of-Expert (Jacobs et al., 1991; Eigen et al., 2013; Ahmed et al., 2016; Riquelme et al., 2021) can also be seen as a way to speed up the inference by selecting a part of the model ("experts") at each time. While our method shares a similar spirit with these works, our model focuses on developing a single model instead of two separate models and still works on the same input resolution. Our "model experts" also have different computation complexity, which allows it to be more adaptive and achieves a better speed-accuracy tradeoff.

There are also some works that focus on the model distillation (Hinton et al., 2015; Touvron et al., 2021; Xiong et al., 2023) as well as model quantization (Dettmers et al., 2022; Xiao et al., 2023; Ma et al., 2024) to speed up the computation. Since our goal is to propose a general model architecture that incorporates computation-intensive and efficient blocks, we argue that our model is complementary to these methods and the speed can be further improved.

## 3 BIG.LITTLE VISION TRANSFORMER

### 3.1 OVERVIEW

The core big.LITTLE module in the bLViT architecture comprises two components: a performance block (P-block) and an efficiency block (E-block). The token processing pipeline is illustrated in Fig. 2. This module processes a sequence of image tokens as input. The importance of each token is predicted beforehand by the prediction layers, allowing for the ranking of tokens based on their importance. The top-K tokens, deemed the most critical, are processed by the P-block, which,

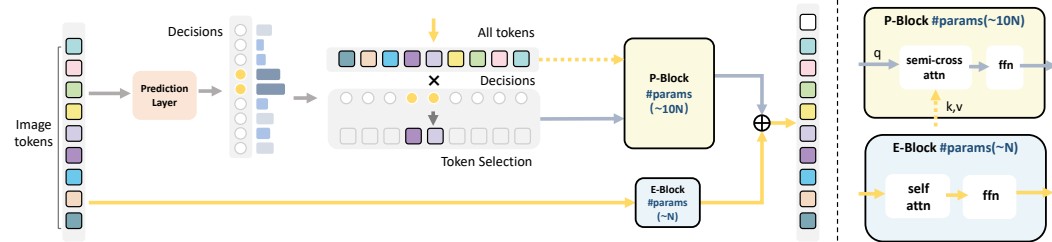

Figure 2: **The Pipeline of big.LITTLE Vision Transformer module.** Left: The module takes the image token sequence as input. The efficiency block (E-Block) updates all tokens with high speed. Then the importance scores from a prediction layer are used to select tokens, where a higher score means more important for the final prediction. The selected tokens are then fed into the performance block (P-Block) with a high capacity. Finally, we fuse the outputs from E-Block and P-Block to form new image representations. Right: P-Block uses semi cross attention to facilitate information interaction between the selected tokens and all tokens, while E-Block is a vanilla ViT block with dimension matching.

though having higher computational capacity, operates at a slower speed. In contrast, the entire token sequence is passed through the E-blocks, which prioritize efficiency, offering faster processing at the cost of lower capacity. The P-block handles the crucial tokens in detail to maintain model performance, while the E-block efficiently updates all tokens to preserve context information at a lower computational cost. The output of P-block and E-block are then fused to form the final output of the big.LITTLE module.

## 3.2 PERFORMANCE-EFFICIENCY BLOCK

---

**Algorithm 1** Pseudo Code of big.LITTLE module in a PyTorch-like style.

---

```
def big_little_forward(x, importance_score, p_ratio):
    # x: input image tokens with shape N x C
    # importance_score: scores from prediction layers
    # p_ratio : the ratio of tokens processed by P_Block

    # top_mask indicates the selected token position
    topk_mask = topk(importance_score, k=p_ratio)

    # Process selected tokens in the dual attention
    x_primary = P_Attention(q=get_primary_tokens(x, topk_mask), kv=x)
    x_secondary = E_Attention(x)
    # fuse the dual data flow, skip connection
    x = x + fuse(x_primary, X_secondary)

    # Process selected tokens in the dual ffn
    x_primary = P_FFN(get_primary_tokens(x, topk_mask))
    x_secondary = E_FFN(x)
    # fuse the dual data flow, skip connection
    x = x + fuse(x_primary, X_secondary)

    return x
```

---

In a big.LITTLE module, the forward function is shown in Algo. 1. We begin with a set of image token $x \in \mathbb{R}^{N \times C}$.

**Token Selection and Routing.** Before the dual blocks, a prediction layer—composed of a linear layer followed by a softmax function—estimates the importance scores of all image tokens, identifying the most crucial tokens for further processing, as shown in Fig. 2. We employ a top-k selection mechanism to select primary tokens based on these importance scores. These selected tokens are then routed to the more computationally intensive P-block. As described in Algo. 1, only a subset

of tokens is processed by the attention and FFN layers of the P-block, while all tokens are updated by the E-block. To enable back-propagation through the prediction layer, we follow (Raposo et al., 2024) by multiplying the scores of selected tokens with the P-block output, formulated as

$$\text{output}_i = (\alpha \cdot s_i + 1) \cdot \text{module}(\text{input}_i),$$

where $s_i$ is the importance score of the token in the $i$-th layer, the module can be the FFN or attention layer in the P-block, and $\alpha$ is a learnable parameter initialized at 0 to stabilize the training process. For simplicity, this part is omitted in the pseudo-code.

**Dimension Matching**   As the E-block and P-block have different model capacities, the hidden dimensions of the representations are inevitably different. To reconcile these differences and accommodate the requirements of both the efficiency and performance blocks, we modify the vallina ViT block for the E-block. Specially, we insert two linear layers in the beginning and ending in the FFN layer to conduct dimension mapping; as for the attention layer, input and output dimensions are modified directly to match the dimension of the main flow. These operations are conducted in E_Attention and E_FFN in the pseudocode.

**Semi-Cross Attention**   In the previous token pruning method, unimportant tokens were directly removed, preventing the remaining tokens from exchanging information with the pruned tokens in the attention layer. To address this issue, we propose a Semi-Cross Attention mechanism for P-blocks. Specifically, in the attention layer of the P-block, we use the primary tokens as queries (q) and all tokens (both selected and unselected) as keys (k) and values (v), instead of only using the same tokens as queries. This allows the primary tokens to still gather information from all tokens, not just from themselves.

**Token Fusion.**   After processing through the dual blocks, the output of the P-block is fused with that of the E-block with the globally updated context. This fusion is performed using a learnable parameter $\gamma$, which adjusts the influence of the tokens on the final output, formulated as

$$x_{\text{fused},i} = \begin{cases} x_{\text{primary},i} + \gamma \cdot x_{\text{secondary},i}, & \text{if } M_i = 1 \\ x_{\text{secondary},i}, & \text{if } M_i = 0 \end{cases}$$

Here, $M$ is a binary mask indicating whether the $i$-th token is a primary token ($M_i = 1$) or not ($M_i = 0$). This ensures that the most significant features are emphasized while maintaining the overall integrity of the data representation.

**Variants of P-E block.**   In practice, the configuration of P-block and E-block can vary depending on the model size, and inner dimensions of both P-block and E-block follow variants of the vanilla ViT block. For instance, we can set the dimensions of the P-block and E-block as those of ViT-Base block and ViT-Tiny block respectively, as the E-block to match ViT-Base performance while saving computation. Here, we adopt a 1:1 stacking ratio of P and E blocks, meaning each layer of image tokens passes through one P-block and one E-block. In models with a larger size, such as a huge-base combination, we might employ a 2:1 stacking ratio or other variations.

**Theoretical Computation Analysis.**   To reduce computational demands, we empirically let the performance blocks process the top $25\%$ most important tokens by default, while the efficiency block updates all tokens, ensuring comprehensive coverage of context information. In this way, our model allocates computational resources for each token adaptively based on its content, leading to a better speed-accuracy tradeoff. We conduct a simple analysis of how much computation we can save: for input with shape $N \times C$, where $N$ is the number of tokens and $C$ is the hidden dimensions of tokens, the computation cost of a vanilla ViT block is $12NC^2 + 2N^2C(4NC^2 + 2N^2C$ is for attention layer and $8NC^2$ is for FFN). A performance block updates $25\%$ tokens with a cost of $4.5NC^2 + 0.5N^2C(2.5NC^2 + 0.5N^2C$ is for semi-cross attention and $2NC^2$ is for FFN) and an efficiency block with $\frac{1}{4}C$ hidden dimensions costs $2NC^2 + 0.5N^2C(NC^2 + 0.5N^2C$ is for attention layer and $NC^2$ is for FFN, which is larger than the result of substituting $\frac{1}{4}C$ into $C$ in vanilla cost because of additional overhead incurred by dimension matching) when processing all tokens. This leads to a total cost of $6.5NC^2 + N^2C$, over $1.84\times$ theoretical speedup for each layer, which could further be higher as the efficiency block becomes smaller.

## 3.3 TRAINING STRATEGY

In practice, we find that naively training a model with big.LITTLE modules may lead to suboptimal performance, possibly due to the high pruning ratio, and we empirically find that feature distillation can improve its performance.

During training, feature distillation is used to transfer knowledge from a pre-trained vallina ViT to our big.LITTLE ViT. By aligning the features learned by the student with those of the teacher, the model can retain critical information even when aggressive pruning is applied. The feature distillation loss is formulated as:

$$\mathcal{L}_{\text{fd}} = \text{cos\_similarity}(\text{feat\_bLViT}, \text{feat\_vallinaViT}),$$

where feat_bLViT represents the feature embeddings from the big.LITTLE model, and feat_vallinaViT represents the embeddings from the pre-trained teacher model. The cosine similarity function ensures that the feature representations of our model are as close as possible to those of the teacher. The total loss used for training combines the supervised loss $\mathcal{L}_{\text{supervised}}$ with the feature distillation loss, weighted by a scalar $\lambda_{\text{fd}}$:

$$\mathcal{L}_{\text{total}} = \mathcal{L}_{\text{supervised}} + \lambda_{\text{fd}} \cdot \mathcal{L}_{\text{fd}}.$$

## 4 EXPERIMENTS

### 4.1 IMPLEMENTATION DETAILS

In our experiments, we employ two variants of the bLViT. In the first variant, we use the ViT-Base as the P-block and the ViT-Tiny as the E-block, denoted as B+T. The model consists of 12 layers as the vanilla ViT-Base, the first layer is the ViT-Base layer where it can see all tokens, and starting from the second layer we start to use big.LITTLE modules, therefore this model consists of 12 P-blocks and 11 E-blocks in total. The prediction layers are used after layers 1, 4, 7 and 10. In the second variant, we test it with a larger model size and use the ViT-Huge as the P-block and the ViT-Base as the E-block, denoted as H+B. This model follows the 32-layer architecture of the standard ViT-Huge, with the first 9 layers exclusively using the ViT-Huge, fully processing all tokens. Starting from the tenth layer, a big.LITTLE module is alternately used in every other layer. In layers without an E-block, only $25\%$ of tokens are updated by the P-block, resulting in a configuration of 32 P-blocks and 12 E-blocks in total. Here, the prediction layers are used after layers 8, 16 and 24.

For models with window attention such as SAM (Kirillov et al., 2023), token selection occurs within each window, ensuring the same number of tokens in different windows, which facilitates parallel computation.

All experiments are conducted on 8 NVIDIA A100 GPUs. $\gamma$ is initialized to $10^{-5}$. $\lambda_{\text{fd}}$ is set to 2.5 by default. AdamW optimizer is applied in the experiment, with learning rate of $5 \times 10^{-4}$ in both sets of tasks.

### 4.2 BASELINES AND EVALUATION METRICS

We compare our method with existing token pruning methods for ViT structure, i.e., AdaViT (Meng et al., 2022), ATS (Fayyaz et al., 2022), A-ViT (Yin et al., 2022), DynamicViT (Rao et al., 2021), Evo-ViT (Xu et al., 2022), E-ViT (Liang et al., 2022), efficient ViT models, i.e., EfficientViT (Liu et al., 2023b), MobileViT (Mehta & Rastegari, 2021), and also include the comparison with vanilla ViT (Touvron et al., 2021). We validate the performance on two tasks including image classification and segment anything task.

**Image Classification.** We choose the vanilla ViT as the baseline. The Top-1 accuracy is employed as the evaluation metric. Three vanilla ViT variants from DeiT were employed. For ATS, A-ViT, DynamicViT, Evo-ViT, E-ViT, and our method, the pretrained weights of DeiT were used for initialization, followed by training on the ImageNet-1K dataset for 300 epochs with a batch size of 1024 on 8 GPUs, and then tested for top-1 accuracy in image classification. The training details followed DeiT (Touvron et al., 2021). For AdaViT, it was initilaized by T2T-ViT (Yuan et al.,

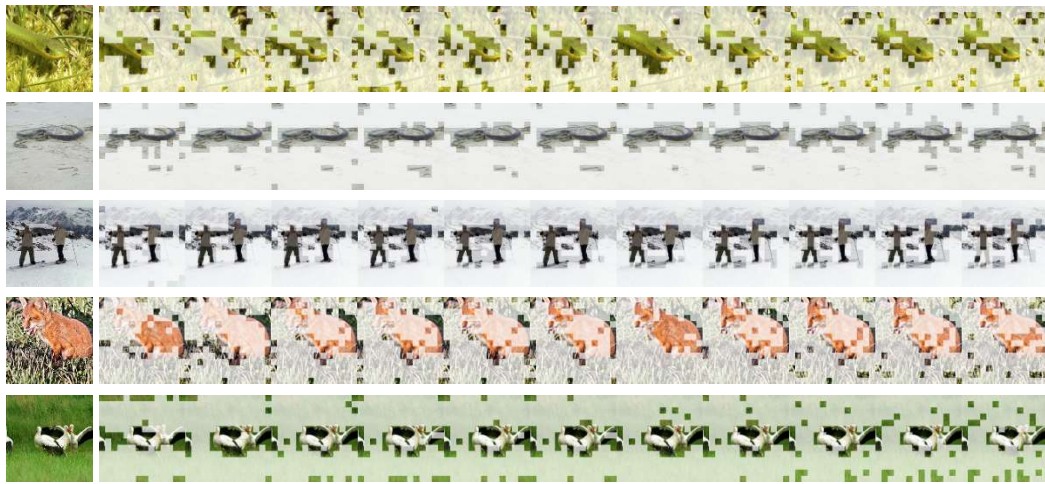

Figure 3: **Token Selection Visualization.** In bLViT, the tokens processed in the high-capacity P-block highlight areas crucial for image classification.

2021), which is marked with an asterisk in Table 1. We adopted multiple settings in some methods. For EfficientViT, we used the models corresponding to resolutions of 224 and 512 under the M5 configuration. DynamicViT utilized two model sizes (base and small), and EViT used two keep ratios of 0.5 and 0.6.

**Segment Anything.** The evaluation is similar to SAM, where segmentation is performed from a single foreground point, a single box, and multiple points. Here, random points are uniformly sampled within the ground truth mask for clicking, and the ground truth box is used as the prompt box. We also conduct zero-shot instance segmentation experiments, following the setting of SAM (Kirillov et al., 2023). Regarding the baseline, vanilla variants of SAM were trained on the complete SA-1B dataset for 2 epochs. For Evo-ViT and E-ViT, two experimental setups were divided: ViT-Base and ViT-Huge. In both setups, the pretrained weights of vanilla SAM were used for initialization. Correspondingly, the big.LITTLE configurations B+T and H+B were used. During training, the models were trained for 10 epochs on $2\%$ of the SA-1B dataset with a batch size of 8. For testing, the LVIS (Gupta et al., 2019) dataset was utilized to evaluate the mask prediction performance of the models, and the COCO dataset was used in zero-shot instance segmentation.

### 4.3 IMAGE CLASSIFICATION

We conducted experiments on the ImageNet-1k classification dataset (Krizhevsky et al., 2012) and report the top-1 accuracy and GFLOPs in Table 1. The results demonstrate that our method achieves the best performance. Specifically, our Base + Tiny bLViT reduces computations by about 50% while outperforming ViT-B. Although methods utilizing light architectures exhibit significantly lower computational costs compared to most efficient ViT approaches, their performance is severely limited by model capacity. In the efficient ViT group, the performance of ATS and A-ViT, both based on ViT-Small, significantly lags behind our model. Our method achieves the best performance and the second-best computational efficiency compared to models based on ViT-Base. Notably, our model is the only one based on ViT-B that surpasses its performance, while other similar models tend to sacrifice performance for reduced computational costs, as illustrated in Fig. 1.

Further, we visualize which tokens pass through the P-block in the 11-layer big.LITTLE module. As illustrated in Fig. 3, after training, the model effectively selects regions critical for image classification to be processed by the high-capacity P-block. This capability highlights the architectural efficiency and targeted processing power of our bLViT.

### 4.4 SEGMENT ANYTHING TASK

With the models trained on SA-1B dataset, we validate them on two types of experiments, as shown in Table 2. We report mIoU under three settings of mask prediction and AP under zero shot instance segmentation, respectively. From the table, one can see that our model largely reduces the

Table 1: **ImageNet classification results.** We report Top-1 accuracy and GFLOPs for vanilla ViT models with different scales, light architectures of ViT and efficient ViTs based on token pruning. Our model achieves better accuracy-computation tradeoff.

| Type | Model | Acc | GFLOPs |
|---|---|---|---|
| Vallina ViT | ViT-T | 72.2 | 1.08 |
| | ViT-S | 79.9 | 4.24 |
| | ViT-B | 81.8 | 16.86 |
| Light architecture | EfficientViT-M5 | 77.1 | 0.52 |
| | EfficientViT-M5-512 | 80.8 | 2.67 |
| | MobileViT-XS | 74.8 | 0.7 |
| Efficient ViT | AdaViT* | 81.1 | 4.0 |
| | ATS-S | 79.7 | 2.9 |
| | A-ViT-S | 80.7 | 3.6 |
| | DynamicViT-S | 79.6 | 3.4 |
| | DynamicViT-B | 81.3 | 11.2 |
| | Evo-ViT-B | 81.2 | 11.30 |
| | EViT-B @ 0.5 | 80.0 | 8.40 |
| | EViT-B @ 0.6 | 81.7 | 9.66 |
| | Ours (B + T) | 81.9 | 9.89 |

computation, reflected in that our B+T version reduces about half of the GLOPs compared with ViT-B. Also, our approach outperforms other accelerating techniques significantly, with the highest performance and also the highest efficiency. Notably, under the testing settings of three points and bounding boxes, our models even surpasses ViT-B and ViT-H respectively. The potential explanation for this phenomenon could be attributed to the signals obtained from both the distillation loss and the supervision loss.

Table 2: **Segment anything results.** Our big.LITTLE not only demonstrate a substantial reduction in computational demand, but also achieve comparable performance, outperforming other acceleration techniques, even baselines.

| Model | 1 Point | 3 Points | Box | Zero shot instance segmentation | GFLOPs |
|---|---|---|---|---|---|
| ViT-B | 53.6 | 65.2 | 76.6 | 40.2 | 372.0 |
| ViT-H | 59.4 | 70.7 | 80.4 | 46.1 | 2736.6 |
| Evo-ViT-B | 39.9 | 61.4 | 71.1 | 30.7 | 266.0 |
| EViT-B @ 0.5 | 40.1 | 60.8 | 70.1 | 32.4 | 216.4 |
| Ours (B + T) | 52.0 | 71.1 | 78.1 | 39.2 | 210.5 |
| Evo-ViT-H | 42.3 | 63.1 | 72.5 | 38.8 | 1840.5 |
| EViT-H @ 0.5 | 40.5 | 62.7 | 72.2 | 39.5 | 1597.2 |
| Ours (H + B) | 58.6 | 72.6 | 81.4 | 45.0 | 1993.3 |

## 4.5 ABLATION STUDY

**Model design.** We conduct ablation studies on the ImageNet classification task to verify our model design choices. Besides the vanilla DeiT-Base model without any token pruning, we also select Evo-ViT with 81.0 Top-1 Accuracy and 50% token pruning ratio as our baseline model and illustrate how we reach our final model design. We can see that, while naively increasing the pruning ratio to 75% and reducing the number of performance blocks (Early Prune) can save the computation and we observe a decent FLOPs reduction, the performance also drops severely. Simply adding the efficiency block (E-Block) can mitigate this issue, but still fall behind the baseline. We then apply prediction layers (Predictor) and semi-cross attention (Semi-CA) to bridge this gap. Then, we leverage the pretrained weights initialization, where the weights of the performance blocks are pretrained without any token pruning. We empirically find this yields better performance. Finally, we

use feature distillation (Feat. Dis.) that are described in § 3.3 during the training process to obtain the best performance.

Table 3: **Ablation studies on ImageNet classification task.** We start from ViT-Base and Evo-ViT-Base and verify our design choice step by step.

| Prune Ratio | Early Prune | E-Block | Predictor | Semi-CA | Pretrain | Feat. Dis. | Acc | GFLOPs |
|---|---|---|---|---|---|---|---|---|
| 0% | | | | | | | 81.8 | 16.8 |
| 50% | | | | | | | 81.0 | 11.3 |
| 75% | | | | | | | 79.0 | 8.5 |
| 75% | ✓ | | | | | | 74.2 | 5.4 |
| 75% | ✓ | ✓ | | | | | 78.7 | 7.9 |
| 75% | ✓ | ✓ | ✓ | | | | 78.8 | 8.0 |
| 75% | ✓ | ✓ | ✓ | ✓ | | | 79.5 | 9.9 |
| 75% | ✓ | ✓ | ✓ | ✓ | ✓ | | 80.8 | 9.9 |
| 75% | ✓ | ✓ | ✓ | ✓ | ✓ | ✓ | 81.9 | 9.9 |

Table 4: **Ablation studies on distillation loss scalar.**     Table 5: **Ablation studies on pruning ratio.**

| $\lambda_{fd}$ | Acc |
|---|---|
| 1.0 | 81.3 |
| 2.5 | 81.9 |
| 5.0 | 81.7 |
| 10.0 | 81.2 |

| Pruning Ratio | Acc |
|---|---|
| 0.5 | 82.3 |
| 0.625 | 82.1 |
| 0.75 | 81.9 |
| 0.875 | 81.2 |

**Distillation loss scalar.**   When using feature distillation loss for model training, the coefficient for this loss needs to be set empirically, as values that are too large or too small can hinder optimal performance. In Table 4, one can observe that $2.5$ is a notable discrete peak value worth adopting.

**Pruning ratio.**   In our model, when entering the P-block, a portion of the tokens will be discarded, and this proportion is referred to as the pruning ratio. Intuitively, the performance tends to decrease as the pruning ratio increases. Therefore, we need to balance the trade-off between model performance and computational efficiency. In Table 5, we can roughly observe that when the pruning ratio is less than $0.75$, the decline in performance becomes less pronounced as the pruning ratio increases; however, beyond this point, the decline becomes noticeably faster. Consequently, we empirically adopt a pruning ratio of $0.75$.

## 5    CONCLUSION

This paper introduces the big.LITTLE Vision Transformer (bLViT), an innovative architecture designed to enhance the efficiency of visual recognition systems. By strategically allocating image tokens between a high-capacity performance block and a speed-optimized efficiency block, this architecture significantly reduces computational demands while maintaining high accuracy. Our experimental results demonstrate that the bLViT not only preserves robust accuracy but also boosts computational efficiency, making it a practical choice for scalable and adaptable AI deployments.

## BROADER IMPACT

Our work aims to improve the inference speed of the vision transformer models. Our model design can allow the vision transformer model to run on cheaper and more energy-efficient hardware at an acceptable speed. It would benefit people without access to expensive hardware and make a positive impact on combating climate change since the inference becomes more efficient. We acknowledge unknown risks can be brought by the development of AI technology; however, the contribution of this paper has no greater risk than any other generic deep-learning paper that studies standard datasets such as ImageNet and MSCOCO.

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
