# OpenReview forum: "big.LITTLE Vision Transformer for Efficient Visual Recognition"
_ICLR.cc/2025/Conference — ICLR 2025 Conference Withdrawn Submission_

### Official Review · Reviewer_B2za · 2024-11-01

**Soundness:** 3
**Presentation:** 4
**Contribution:** 2
**Rating:** 3
**Confidence:** 5

**Summary:**

This paper introduces a new architecture named big.LITTLE ViT to enhance the visual recognition tasks. The key innovation is a dynamic inference mechanism that allocates image tokens to P-block or E-block based on their importance. Comprehensive experiments show that the proposed method achieves a competitive balance between computational speed and accuracy.

**Strengths:**

1. The paper is well written and organized.
2. The proposed P-Block and E-Block are reasonable, leading to less computational costs without compromising accuracy.
3. The experiments on image classification and SAM somehow verifies the effectiveness of the proposed method.

**Weaknesses:**

1. The actual speed-up of the proposed method (on GPU/CPU) should be included compared to the previous method.
2. Some related works are missed about the efficient architecture of SAM[1][2]. I think the proposed method should be compared with them.
3. The characters in the figures (such as Figure1) should be larger for the convenience of the readers.

[1]Tinysam: Pushing the envelope for efficient segment anything model.
[2]Efficientvit-sam: Accelerated segment anything model without performance loss.

**Questions:**

See weaknesses

---

### Official Review · Reviewer_7Mwd · 2024-11-02

**Soundness:** 3
**Presentation:** 3
**Contribution:** 2
**Rating:** 5
**Confidence:** 4

**Summary:**

The manuscript introduces the big.LITTLE Vision Transformer (bLViT), designed to enhance efficiency in visual recognition tasks. The architecture includes a module of “big” performance blocks and the other module of “LITTLE” efficiency blocks, in which performance blocks focus on processing a number of selected “important” tokens, and efficiency blocks handle all tokens more rapidly, particularly those deemed less significant. Experimental results on tasks including classification and segmentation demonstrate the method's effectiveness, achieving a reduction in theoretical computational complexity together with performance optimization.

**Strengths:**

(1)	The manuscript addresses the challenges associated with sub-optimal performance and computational complexity due to the omission of unrepresentative tokens in vision transformers. The motivation for the proposed approach is clear, and the methodology is intuitive and accessible. The integration of performance and efficiency blocks is straightforward and effective.

(2)	The manuscript is well-organized, exhibiting clear logic and structure, which facilitates reader’s understanding.

**Weaknesses:**

(1)	While the proposed method shows notable improvement over the baseline, it does not demonstrate significant advantages in terms of performance or computational complexity relative to other SOTA approaches. This could limit the technical merits and impacts of the work.

(2)	The manuscript would benefit from a more in-depth exploration and discussion of the proposed methods. Additional experiments on analyzing which tokens are utilized and how to optimize the use of important tokens are suggested to conduct in order to enhance the manuscript's significance and inspire further research.

(3)	The analysis part focuses solely on theoretical FLOPs without a corresponding comparison of actual inference speeds. It is important to note that lower theoretical FLOPs do not necessarily generate faster inference speeds in practice.

(4)	The introduction of additional hyperparameters in the proposed method is expected to be clarified and further explained, as tuning these parameters may incur substantial time costs, potentially undermining the intended reduction in computational complexity for vision transformers.

**Questions:**

(1)	Is it able to provide a detailed derivation of the theoretical calculation analysis presented in the manuscript? This will definitely help to clarify the derivation steps.

(2)	What specific structural designs are employed in the performance and efficiency blocks? Additionally, what rationale underpins the parameterization of the performance blocks at 10N compared to N for the efficiency blocks?

(3)	Figure 3 indicates that the important tokens identified by different layers show the minimal variation. Is it necessary to process tokens at each layer? What potential impacts might arise if this layer-wise processing is omitted?

(4)	Beyond enhancing high-level tasks such as classification and segmentation, how effective is the proposed method for low-level tasks, such as super-resolution and de-noising? Insights into its applicability in these contexts would be valuable.

---

### Official Review · Reviewer_ufPj · 2024-11-03

**Soundness:** 2
**Presentation:** 3
**Contribution:** 2
**Rating:** 3
**Confidence:** 4

**Summary:**

This paper presented a new network architecture that balances the performance and computations by leveraging two-stream architecture. One stream takes the important tokens and processing it with more parameters while using a light-weight stream to process all tokens; then, combining the results. The results show that the proposed methods outperform other SOTA methods that aim to improve the efficiency of ViT.

**Strengths:**

1. As pointed by the authors, improving the efficiency of ViT is an important topic, and this work proposed a new architecture to resolve it.
2. The ablation study in model design is comprehensive to discuss the contributions of each component.

**Weaknesses:**

1. The biggest concern is about the need of distillation from a vanilla model, which makes the comparison to others are unfair, based on Table 3, without distillation, it only achieves 80.8, which is worse than EViT-B@0.6 in Table 1. Moreover, what vanilla model is used for the distillation is not described.

2. Following by point 1, for the results in Table 2 and the description at line 404, it shows that the significane of dislliation, this makes the comparison in Table 2 not meaningful; or on the other hand, why the distillation can help significantly in SAM.

3. Missing important reference when comparing to the efficient ViT [1]. This paper proposed better and faster architecture for both image classification and segmentation.

[1] EfficientViT: Multi-Scale Linear Attention for High-Resolution Dense Prediction, ICCV 2023

**Questions:**

See Weaknesses.

1. As the authors discussed the FPS for those heavy model at line 045, what is the FPS of the proposed architecture compared to others?

2. In Table 1, what is the number of parameters comparison?

3. Given the gamma is only 0.0001, do it really provide any information to the tokens processed by P-block?

4. The description in Algo 1 is inconsistent to Fig. 2, as there is a fuse operation right after attention and before ffn, but it is not in Fig. 2.

---

### Official Review · Reviewer_2ooW · 2024-11-05

**Soundness:** 3
**Presentation:** 2
**Contribution:** 2
**Rating:** 5
**Confidence:** 4

**Summary:**

This paper introduces a dual-transformer architecture with big and little performances blocks to dynamic routing the tokens for different computational complexity.
Essential tokens and less critical tokens are processed by high performance big model and more efficient little model respectively.
This effectively reduce the computational load while preserving the model performances.

**Strengths:**

1. P-E block dual transformer design is interesting to leverage the varying importance of different tokens.
2. Semi-Cross Attention mechanism allows important tokens to still gather information from all tokens.

**Weaknesses:**

1. It's unclear how the FLOPs reduction translate to latency/FPS improvement.
2. It's not clear how the duel transformer modules affects self attention mechanism.
3. The proposed method relies on the distillation from pre-trained models which adds training complexity.

**Questions:**

1. Could the author add some result comparing the latency/FPS improvement over the current state of the art models?
2. How would the dual transformer modules affect the attention mechanism?
3. What's the design principle for the current P-E blocks? Is 1:1 stack ratio the optimal choice?
4. How's the performance compared to some hybrid architectures such as FastViT: A Fast Hybrid Vision Transformer using Structural Reparameterization?
5. How much impact does token fusion have on the overall model performances?

---

### Note · Authors · 2024-11-15

I have read and agree with the venue's withdrawal policy on behalf of myself and my co-authors.